# Efficacy and Safety of Virtual Reality-Based Versus Traditional Emotion-to-Emotion Therapy for Treatment of Hwa-Byung: A Protocol for a Single-Center, Randomized, Assessor-Blind, Parallel-Group Clinical Trial

**DOI:** 10.3390/healthcare12232407

**Published:** 2024-11-30

**Authors:** Hye Jeong Kook, Dong Hoon Kang, Yang Chun Park, Nam Kwen Kim, Hyung Won Kang, In Chul Jung

**Affiliations:** 1Department of Oriental Neuropsychiatry, College of Korean Medicine, Daejeon University, Daejeon 34520, Republic of Korea; khjehhs@gmail.com (H.J.K.); 20234141@edu.dju.ac.kr (D.H.K.); 2Department of Internal Medicine, College of Korean Medicine, Daejeon University, Daejeon 34520, Republic of Korea; omdpyc@dju.kr; 3Center for Big Data & Comparative Effectiveness Research, Economic Evaluation in Health and Medicine, Pusan National University, Busan 43241, Republic of Korea; drkim@pusan.ac.kr; 4Department of Korean Neuropsychiatry Medicine, College of Korean Medicine, Wonkwang University; Iksan-si 54538, Republic of Korea; dskhw@wku.ac.kr

**Keywords:** Hwa-Byung, emotional-to-emotion therapy, virtual reality, digital therapeutics, psychotherapy, randomized clinical trial

## Abstract

Background: Hwa-Byung is a culturally bound syndrome prevalent in Korea, characterized by intense emotional distress and physical symptoms related to suppressed anger. Patients frequently report experiencing chest tightness, heat sensations, and globus pharyngeus (the sensation of a lump in the throat). These physical symptoms often accompany psychological distress and can exacerbate the sense of frustration and helplessness associated with the condition. The distinctive presentation of these symptoms highlights the necessity for therapeutic interventions that address both the emotional and physical aspects of Hwa-Byung. Conventional therapeutic methods, such as Emotion-to-Emotion Therapy (ETE), have shown efficacy in treating this condition. This study aims to assess the efficacy of Virtual Reality-based Emotion-to-Emotion Therapy (VR-based ETE) compared to conventional ETE, utilizing immersive digital platforms to enhance therapeutic engagement. Methods: This single-center, randomized, assessor-blind, parallel-group clinical trial will enroll 96 participants, evenly divided into two experimental groups and one control group. Over eight weeks, each participant will undergo 12 intervention sessions. Experimental group 1 will receive conventional ETE, experimental group 2 will receive VR-based ETE, and the control group will receive Hwa-Byung management training materials. The primary outcome will be the change in symptoms, measured by the Likert Scale for Major Symptoms of Hwa-Byung from baseline to 8 weeks. Secondary outcomes will include psychometric scales and physiological measures such as the core seven-emotions inventory short form, physical health questionnaire, stress response index, Beck depression inventory, state-trait anger expression inventory, state-trait anxiety inventory, functional near-infrared spectroscopy, and heart rate variability. For economic efficiency assessment, quality-adjusted life-years will be the primary outcome using the EuroQol-5 dimension, and the secondary outcome will be using the EuroQol visual analog scale. Discussion: ETE is a recognized oriental psychotherapy that enhances symptom management, self-regulation, and stress coping. VR is expected to deepen treatment immersion. By combining these strengths, VR-ETE may further improve emotion regulation and alleviate psychosomatic symptoms. If successful, this study will not only advance the treatment of Hwa-Byung but also contribute to the modernization of traditional Korean medicine through the integration of digital therapies.

## 1. Introduction

Emotion dysregulation is a key factor in many psychosomatic and psychiatric disorders globally [1]. For example, anxiety disorders often involve excessive, uncontrollable worry and fear; depression is marked by intense sadness, irritability, and frustration that are hard to regulate; and anger disorders feature frequent, intense outbursts disproportionate to their triggers, indicating poor emotional control [2,3,4]. Emotion dysregulation can manifest in various forms, contributing to the overall symptomatology and impairment in each condition [5]. It contributes to the development of conditions like Hwa-Byung [6]. Hwa-Byung, also known as “anger syndrome”, is attributed to the suppression of anger within a Korean cultural context [7]. It affects approximately 4.1–13.3% of South Korea’s general population, with a higher prevalence among middle-aged women [8,9,10]. Patients frequently endure prolonged periods of anger, displaying symptoms such as chest discomfort, heat sensations, heartburn, globus pharyngeal, and frequent feelings of unfairness and resentment, which are major diagnostic criteria for Hwa-Byung [11,12]. The trajectory of Hwa-Byung typically progresses from prolonged psychological stress to physical symptoms such as chest tightness and heat sensations, reflecting a build-up of suppressed emotions over time [13]. Hwa-Byung is often rooted in sociocultural expectations in Korea, where expressing anger directly is sometimes discouraged, especially within familial and social settings [13]. Research indicates that the inability to express anger, lack of social support, spousal criticism, or interpersonal hypersensitivity increases the likelihood of developing Hwa-Byung [14]. Although Hwa-Byung is culturally specific to Korea, the underlying emotional dysregulation is universal and mirrors psychosomatic conditions seen globally, such as anxiety and depression [6]. Similar culturally specific syndromes, such as taijin kyofusho in Japan and ataque de nervios in Latin American populations, have been studied to understand the psychological and therapeutic approaches tailored to culturally bound expressions of distress [15]. These studies underscore the importance of a cultural lens in understanding psychological issues, as an individual’s history within their cultural context shapes symptom presentation and treatment outcomes, highlighting the need for culturally adaptive interventions [13,15].

Emotion-to-Emotion Therapy (ETE) is a psychotherapy based on the theory of the five elements, the basic principle of traditional Korean medicine, which treats mental and physical ailments by alleviating pathological emotions and is recommended in clinical practice guidelines for the treatment of Hwa-Byung [16]. The five emotions of anger, joy, thought, sadness and melancholy, fear and surprise are assigned to each of the five elements: Tree, Fire, Earth, Metal, and Water, dynamically influencing each other [17]. The treatment is based on controlling this dynamic relationship, analyzing the patient’s emotions and feelings, and using the corresponding emotions and feelings in the Five Elements to help the patient achieve a stable state [17]. Traditional Korean medicine psychotherapy employs a holistic approach aimed at balancing physical and emotional states through methods such as meditation and mind–body exercises [18]. ETE applies these principles to address the psychosomatic symptoms of Hwa-Byung by helping patients identify and regulate core emotions, such as anger, resentment, and sorrow, thus restoring emotional balance [12]. This therapy also promotes self-compassion and compassion toward others, facilitating the transition from emotional distress to resilience and enabling patients to manage anger, adapt to stress, and resume healthy functioning [13].

Virtual Reality (VR) therapy has been demonstrated to be an efficacious treatment for a range of mental health disorders, alleviating patients’ stress [19,20]. The COVID-19 pandemic has led to an increase in in-person interactions in all areas of life with these interactions becoming increasingly crucial in the field of psychotherapy [21,22]. VR is well suited for non-face-to-face interaction and provides a guaranteed, reproducible, controllable, and safe environment for therapy and training [23,24].

There is evidence that VR can enhance psychological treatments for a variety of neuropsychiatric diseases, such as anxiety disorders [19,25,26]. However, there is a paucity of research investigating its potential for the treatment of Hwa-Byung. Thereby, the integration of Virtual Reality (VR) therapy into the treatment of Hwa-Byung represents a promising avenue for enabling patients to safely confront stressors and emotional triggers within a structured, therapeutic setting. A potential VR intervention design is illustrated in Figure 1. The proposed intervention design for Hwa-Byung treatment through VR comprises an initial assessment, immersive exposure to VR scenarios pertinent to emotional regulation, real-time biofeedback, and guided reflection sessions. These steps provide a structured approach for addressing the emotional and physiological symptoms of Hwa-Byung, thereby enhancing the therapeutic efficacy of VR.

Patients suffering from anger, depression, and anxiety associated with Hwa-Byung are prone to emotional instability, potentially leading to comorbid somatoform disorders. This vulnerability complicates their ability to manage daily tasks and interpersonal relationships, thereby increasing the frequency of medical treatments and associated costs [14]. Indeed, statistical data reveal that the number of Hwa-Byung cases treated in Korea increased by 10.5% from 2015 (12,592 patients) to 2019 (14,064 patients), with medical expenses rising by 48.2% over the same period [27]. Although Selective Serotonin Reuptake Inhibitors (SSRIs) have been employed to manage symptoms [28], comprehensive clinical studies are lacking. A retrospective study on the emotional characteristics of Hwa-Byung patients showed a dynamic change over time in emotions such as sorrow, underscoring the need for a therapy that considers these unique emotional traits [29]. Traditional Korean medicine psychotherapy addresses psychological conditions holistically, combining methods like herbal medicine, acupuncture, and emotional regulation techniques [30]. Despite some applications of traditional Korean medicine psychotherapy for Hwa-Byung [31], there is a lack of standardized protocols, especially those that address emotions per se. Therefore, further research is needed to confirm the efficacy of emotion-to-emotion therapy focused on specific emotions and to determine the most appropriate method of application for Hwa-Byung.

Quality-Adjusted Life-Year (QALY) is a widely used measure in healthcare cost-effectiveness analysis, combining quality and length of life to evaluate treatment value [32]. QALY assessments are commonly employed in countries like the UK and the US to guide resource allocation by balancing costs with patient benefits [33,34]. For VR-based therapies, QALY offers insights into cost effectiveness compared to traditional methods, showing that high QALY scores support the economic feasibility of innovative treatments like VR-based ETE, which may help reduce the long-term burden of conditions such as Hwa-Byung [35].

This study aims to determine the safety, effectiveness, and cost effectiveness of virtual reality-based emotion-to-emotion therapy compared to traditional emotion-to-emotion therapy for Hwa-Byung.

## 2. Methods/Design

### 2.1. Study Design

This study is a single center, randomized, assessor-blind, parallel-group clinical trial. This clinical trial will be conducted at Daejeon Korean Medicine Hospital of Daejeon University. This protocol (version 1.2, October 2022) complies with the Standard Protocol Recommendations for Interventional Trials (SPIRIT) guidelines [36]. The SPIRIT schedule of enrollment, interventions, and assessments is shown in Figure 2. The flowchart of this study is shown in Figure 3. This study was registered in the Clinical Research Information Service of the Republic of Korea (Trial Registration Number: KCT0007871, registered on 3 November 2022, available at [https://cris.nih.go.kr/cris/search/detailSearch.do/27510 (accessed on 25 May 2024)].

### 2.2. Study Procedure

Before conducting this clinical trial, the investigator will explain the “informed consent form” to the participants and receive written consent from the participants for the clinical trial based on their free will after ensuring that they have sufficiently understood what is required of them. The subjects will be given a screening code after obtaining their consent:

# DJ-S-ZZZ [DJ: Daejeon Korean Medicine Hospital of Daejeon University, S: initial screening, ZZZ: Serial number [001, 002~]] [ex. DJ-S-012: 12th applicant for screening of Daejeon Korean Medicine Hospital of Daejeon University].

The eligible subjects who meet the inclusion and exclusion criteria will be selected based on a demographic survey, vital signs (with systolic blood pressure, pulse rate, and body temperature falling within normal ranges), medical history (including chief complaint, onset, motif, past history, smoking, and alcohol ingestion to exclude candidates with cardiovascular disease, severe respiratory conditions, or significant mental health disorders unrelated to Hwa-Byung), chest X-ray, electrocardiography (EKG), laboratory tests, Beck Depression Inventory (BDI) (excluding subjects who score 2 or higher on item 9 to minimize suicide risk), and Hwa-Byung Diagnostic Interview Schedule (HBDIS) at screening. The included subjects will be assigned an identification code:

# DJ-E-ZZZ [DJ: Daejeon Korean Medicine Hospital of Daejeon University, E: initial enrollment, ZZZ: Serial number [001, 002~]] [ex. DJ-E-014: 14th registered subjects of Daejeon Korean Medicine Hospital of Daejeon University].

Screening tests should be conducted within ten days prior to visit 1 (baseline visit) and visit 13 should be conducted within 16 weeks (±7 days). At visit 1, the investigator will check the results of the screening test (including chest X-ray and laboratory tests), physical examination, and changes in medical history and concomitant medication between the screening and visit 1. The subjects will be randomly assigned to one of three groups: experimental group 1, experimental group 2, or the control group, and each subject will be provided with a unique identification code for tracking and analysis purposes. Before the intervention, subjects will complete the Likert Scale for Major Symptoms of Hwa-Byung (HB-M), The Core Seven-Emotions Inventory Short Form (CSEI-S), Patient Health Questionnaire-15 (PHQ-15), Stress Response Index (SRI), State-Trait Anxiety Inventory (STAI), State-Trait Anger Expression Inventory (STAXI), functional Near-Infrared Spectroscopy (fNIRS) (device name: NS1-H20AM), Heart Rate Variability (HRV), efficacy analysis (EQ-5D, EQ-VAS), and cost analysis. Experimental group 1 will be treated with ETE, and experimental group 2 will be treated with VR-based ETE, while the control group will receive Hwa-Byung management training materials only. At visits 1–12, the investigator will check for adverse effects and changes in medical history and concomitant drugs and patient compliance; subjects will receive ETE or VR-based ETE according to the corresponding intervention schedule at each visit. Tests and assessments will be conducted according to the following schedule: visit 8 (efficacy analysis; EQ-5D, EQ-VAS, cost analysis); visit 12 (HB-M, CSEI-S, PHQ-15, SRI, BDI, STAXI, STAI, fNIRS, HRV, efficacy assessment; EQ-5D, EQ-VAS, cost analysis, laboratory tests); and visit 13 (follow-up visit; HB-M, CSEI-S, PHQ-15, SRI, BDI, STAXI, STAI, efficacy assessment; EQ-5D, EQ-VAS, cost analysis, physical examination, adverse effects, changes in medical history and concomitant drugs).

Additional visits may be made at any time other than the scheduled visit when deemed necessary at the request of the subject or discretion of the researcher. If a subject visits on an unscheduled day, adverse reactions, concomitant medications, results of tests performed, and subsequent medical treatment must be recorded in the case record. Even in the case of subjects who drop out, laboratory tests, pregnancy diagnostic tests, and efficacy evaluation tests can be performed during additional visits.

### 2.3. Clinical Assessments

In this study, the clinical assessments are as follows: vital signs (blood pressure, pulse rate, body temperature); laboratory tests (complete blood cell count: hemoglobin, hematocrit, red blood cell count, white blood cell count, platelet count, erythrocyte sedimentation rate; blood chemistry test: total protein, albumin, aspartate transaminase, alanine transaminase, gamma-glutamyl transpeptidase, total bilirubin, blood urea nitrogen, creatinine, glucose, Na+, K+, Cl−, free thyroxine, thyroid-stimulating hormone; urine analysis: specific gravity, pH, erythrocyte, leukocyte, nitrite, protein, glucose, ketone, urobilinogen, bilirubin, microscopy); pregnancy test (urine human chorionic gonadotropin, only for fertile women to ensure that the results are negative, but for menstruation will be conducted in visit 1); EKG (12-lead electrocardiography); chest X-ray; HB-M; CSEI-S; PHQ-15; SRI; BDI; STAXI; STAI; fNIRS; HRV; EQ-5D; EQ-VAS; and cost analysis.

### 2.4. Study Populations

#### 2.4.1. Inclusion Criteria


(1)Aged 19 to 65 years;(2)Diagnosed with Hwa-Byung through HBDIS(3)Participants who voluntarily decided to participate and signed the consent form.


#### 2.4.2. Exclusion Criteria


(1)Participants with risk of suicide (scoring 2 or more points in BDI question 9, with specific suicidal thought or plan);(2)Participants with a history or present illness of severe psychiatric disorders (e.g., hallucination, delusion, etc., will be judged by a specialist in oriental neuropsychiatry or a resident who has been educated by a specialist in oriental neuropsychiatry and has clinical experience in neuropsychiatry of traditional Korean medicine for over 1 year) taking antipsychotic drugs to treat psychosis or receiving psychosis-related treatment;(3)Participants who have received Korean medical treatment or psychiatric treatment to improve symptoms of Hwa-Byung within the previous four weeks;(4)Seriously unstable medical conditions (the investigator will decide based on laboratory test, vital sign, etc.);(5)Participants with hyperthyroidism or hypothyroidism (over 1.5 times the upper reference limit or under 0.75 times the lower reference limit of TSH, Free T4, or taking medication due to hyperthyroidism or hypothyroidism);(6)Participants who are thought to be not appropriate to participate in this trial by the investigator.


#### 2.4.3. Sample Size

The statistical hypothesis test of the evaluation variable is two-sided, and the significance level will be set at 5%. The type 2 error (β) will be set to 0.2, and the power will be maintained at 80%. The ratio of the experimental 1, experimental 2, and control groups should be 1:1:1. We expect the HB-M reduction in the control group and experimental group to be 2.52 and 6.84, respectively, and pooled standard deviation 5.178 based on previous studies conducted with a design similar to this study [37]. Therefore, the required sample size for each group is expected to be “2 × 6.9^2^ × (1.96 + 0.84)^2^/(7.8 − 2.8)2 ≃ 30/group ”. In addition, considering a dropout rate of 25%, we decided to register 32 participants per group, for a total of 96 participants.

#### 2.4.4. Randomization and Blinding

The people in charge of other assignments (or independent statisticians) who are not involved in conducting and evaluating this clinical trial will generate a random assignment list in a specific and reproducible manner. Randomization of experimental group 1, group 2, and the control group will be performed with a block randomization method in a 1:1:1 ratio. The random assignment table will be sealed and kept in a way that can be checked for unsealing, managed separately by the research manager in the presence of the participants. Each group will be randomly assigned with the same probability of each individual being selected using the statistical program StataMP 16 program (StataCorp LLC, 4905 Lakeway Drive College Station, TX, USA). The investigators and assessors will be separated so that the assessors will not know what kind of treatment the subjects have received. It is not possible to maintain investigator blindness.

### 2.5. Interventions

The ETE group (experimental group 1) and VR-based ETE group (experimental group 2) will receive ETE two times per week for the first four weeks and once a week for the next four weeks (a total of 12 times). The daily management group (control group) will receive Hwa-Byung management training materials and wait for eight weeks living a daily life. Hwa-Byung management training materials will be the same for the experimental and control groups.

ETE will be performed by a specialist in oriental neuropsychiatry or a resident who has been educated by a specialist in oriental neuropsychiatry and has over one year of clinical experience in neuropsychiatry of traditional Korean medicine.

We will follow the ETE protocol [17]. It consists of five stages: (1) assessment of emotion and physical symptoms (CSEI-S, Mentalizing the Room of Mind (MRM), SUDS); (2) analyze the relationship between emotion and physical symptoms, analyzing emotion and sensation with oriental neuropsychiatric perspective, and finding core emotion; (3) training—depending on the patient’s emotional state, choose from exposure mindfulness meditation, breathing meditation, somatic mindfulness meditation, upper danjeon meditation, joy meditation, lower danjeon meditation, middle danjeon meditation, resource mindfulness meditation, and integrated-triple danjeon meditation; (4) assessment of emotion and physical symptoms after intervention (CSEI-S, MRM, SUDS); and (5) managing emotion education.

In the analyzing stage, it is essential to identify the core emotions of the participants, as these will inform the selection of the most appropriate mindfulness meditation types for the training stage. Based on the identified core emotion, a targeted objective and desired emotional response will be established to enhance the effectiveness of the meditation practice. For example, to address emotions like fear and surprise, breathing meditation will be used to enhance concentration, somatic mindfulness meditation to increase sensory awareness, and upper danjeon meditation to foster insight. For sadness and depression, joy meditation will be employed to encourage positive emotional responses, and lower danjeon meditation to explore life meaning by engaging emotions beyond sadness. To address anger, middle danjeon meditation will be used to cultivate compassion, and resource mindfulness meditation to elicit alternative emotional responses. Additionally, integrated-triple danjeon meditation will be applied to integrate and harmonize emotions, supporting emotional stability.

VR-based ETE will follow the above ETE protocol using head-mounted displays (HMDs) or virtual reality software implemented with computers and project sets. The VR software (Mentalcare-PV02), developed by the company Joygram specifically for this study, is installed and run on an HP Victus Desktop 15L TH02-0001kr (R5-5600G/RTX3060Ti/32GB/1TB) (manufactured by HP Inc. in China, Brazil, India, or Mexico), and uses the Oculus Quest 2 (manufactured by META (formerly Facebook) in China) as the primary VR headset. With this VR setting, subjects will conduct training to stabilize their emotions with opposing emotions for problem emotions in a virtual environment that is not limited to the clinical environment.

### 2.6. Criteria for Concomitant Drugs

#### 2.6.1. Possible Drugs

It is permissible if a dose of psychiatric drugs (e.g., anti-depressants, anti-anxiety medication, tranquilizers, and sleeping pills) is constant for 4 weeks before participating in the trial. However, when taking drugs, the history of the drug is investigated at the time of the visit, and if there is a change in the type and dose of the drug, it is used as a side effect variable or a correction variable. Drugs for transient care of other diseases will be medicated after confirmation by the researcher. When concomitant drugs, including drugs for other diseases or adverse events, are administered, information about the drugs, including name, purpose, dose, and duration, will be recorded in a progress note.

#### 2.6.2. Prohibited Drugs

There are no specific drugs prohibited from concomitant use, but the drugs that may affect results can be prohibited at the discretion of the researchers’ decision (e.g., starting with new medications, such as anti-depressants, anti-anxiety drugs, anti-psychotics, corticosteroids, female hormones, L-dopa, digitalis, bromide, cyclosporin, disulfiram, isoniazid, and yohimbine, during the clinical trial). Participants will be required to maintain a stable treatment regimen throughout the study period. Initiation of any new treatments, including additional medications or therapeutic interventions outside of the ETE, will result in withdrawal from the study. This criterion is intended to isolate the effects of the ETE intervention on Hwa-Byung symptoms and prevent confounding results from other treatments.

### 2.7. Adverse Event Report

The investigator will report within 24 h any severe adverse events to the Principal Investigator (PI) regardless of whether they are related to the intervention during the trial. In addition, events considered serious by the investigator or that suggest significant risks, contraindications, side effects, or precautions that may be associated with the intervention will be recorded in the progress note as serious adverse events. In this study, the following are considered as severe adverse events: death: life-threatening events, the need for hospital admission or to extend the admission period, events resulting in continuous or severe disability or impairment of function, deformity or abnormality of the fetus.

When severe adverse events occur, the PI will notify the client (Daejeon Korean Medicine Hospital of Daejeon University) immediately and provide within five days an additional report containing detailed information. The trial should be halted until further instructions are provided. The client should promptly report any severe or unexpected adverse events including onset, severity, treatment taken, progress, or casualty to other relevant investigators, the Institutional Review Board (IRB), and the Director of the Ministry of Food and Drug Safety.

### 2.8. Criteria for Discontinuation or Dropout

#### 2.8.1. Discontinuation

If an adverse event or side effect occurs that negatively impacts either the safety of the subjects or the progression of the clinical trial, the ongoing trial should be discontinued. However, safety assessments related to the adverse event or side effect will still be conducted to ensure thorough evaluation and monitoring. The PI and the client will discuss the safety of the intervention and decide whether to proceed or discontinue the trial, and the decision is reported to the IRB and the Ministry of Food and Drug Safety.

#### 2.8.2. Dropout

Subjects will be dropped if they fail to complete the trial due to adverse events or other reasons. The investigator can stop the interventions and tests and drop the subjects from the trial, or the subjects can always drop out of the trial with their own free will. The subjects may be dropped in the following cases: severe adverse events occurred to the subjects; difficulty proceeding with the trial due to adverse events; discovering systemic disease that was not detected in the pre-intervention examination; the subject or a legal representative of the subject requests to stop the trial due to unsatisfaction; the subject does not comply with the instructions of the investigator; the subject withdraws consent to participate in the trial; the subject cannot be tracked to follow-up; the subject is prescribed a treatment that can affect the study result, without the direction or consent of the investigator during the trial period or follow-up period; progression of the trial is considered inappropriate by the investigator.

### 2.9. Compliance Assessment

Compliance assessment will be performed as follows:$$Compliance\left(\%\right)=\frac{Number\;of\;performed\;interventions}{Number\;of\;planned\;interventions}\times 100$$

Compliance (%) during the trial should be at least 75%. If the compliance (%) of the subjects is less than 75%, it is considered poor and is excluded from the Per-Protocol (PP) analysis.

### 2.10. Statistical Analysis

Efficacy analysis will be evaluated primarily based on the Full Analysis Set (FAS) principle and secondarily according to the Per-Protocol (PP) principle of the primary outcome. Missing values will be imputed appropriately according to why they occur. Subjects are excluded from FAS analysis in following cases: failure to meet eligible criteria, no intervention specified in the clinical study plan has been received, no data have been collected because they have never been evaluated since randomization. Subjects will be excluded from the per-protocol (PP) analysis if they meet any of the following conditions: 1. Dropped out during the trial. 2. Violated the inclusion or exclusion criteria. 3. Had less than 75% compliance with the overall intervention. Additionally, other serious violations of the study protocol may also be considered as grounds for exclusion from the PP analysis. The demographic data of the study subjects included in this trial and baseline clinical history data will be presented with mean and standard deviation (SD) for continuous data for each group and frequency and percentage for categorical data. For each group, an independent sample t-test or Wilcoxon rank sum test will be performed depending on normality in the case of continuous variables, and for categorical variables, a Pearson chi-square test or Fisher’s practice test will be performed.

### 2.11. Efficacy Assessment

#### 2.11.1. Primary Outcome

The primary outcome will be assessed based on the change in the HB-M score from baseline to 8 weeks. The total HB-M score of subjects who received at least one intervention and for whom a score was measured at least once before the trial or after the intervention will be subjected to analysis. If there is a missing HB-M score, after diagnosing the amount and mechanism of missing values, FAS analysis will be performed by selecting an appropriate imputation method. The change in the HB-M score between the two groups will be verified using an independent t-test. We will use analysis of covariance (ANCOVA) if there is a significant difference in the baseline, and multiple regression analysis if there is a significant difference in the other underlying variables.

#### 2.11.2. Secondary Outcome

The secondary outcome will be assessed by changes in the HB-M score from baseline to 16 weeks, changes in the CSEI-S, PHQ-15, SRI, BDI, STAXI, and STAI from baseline to 8 and 16 weeks, and changes in the fNIRS and HRV from baseline to 8 weeks.

CSEI-S measures the core emotions of the patient in a clinical setting [38]. PHQ-15 assesses depression and physical symptoms [39]. SRI measures the degree of stress response [40]. BDI measures the current severity of depression, regardless of psychiatric diagnosis [41]. STAXI measures anger experiences and forms of anger expression [42]. STAI measures state anxiety and trait anxiety [43]. fNIRS shows prefrontal cortex activation which is prominent during both emotional processing and regulation [44]. HRV measures heart rate variability consistent with autonomic nervous system functional status [45].

We will use ANCOVA or baseline-variables-corrected multiple regression analysis for the efficacy assessment of the CSEI-S, PHQ-15, SRI, BDI, STAXI, STAI, fNIRS, and HRV variables. The baseline score will be used as a covariate, in which case a PP analysis will be performed without handling missing values.

### 2.12. Safety Assessment

Changes in laboratory test results before and after the trial will be clinically evaluated. All adverse events during the trial will be listed with a detailed descriptive explanation. Data will be collected through patient self-reporting and researcher observation. The frequency of adverse events related to and unrelated to interventions will be recorded and presented as descriptive statistics. A list of adverse reactions, including frequency, expression rate, definite occurrence time, occurrence rate, severity, and intervention casualty, will be presented with graphs, if necessary. If statistical analysis is required, the paired t-test, McNemar test, analysis of variance [ANOVA], t-test, chi-square test, and/or Fisher’s exact test will be performed according to the characteristics of the variables and the purpose of the statistical evaluation.

### 2.13. Economic Efficiency Assessment

An economic evaluation alongside clinical trial will be performed to check the cost effectiveness among the interventions in this trial. The primary economic endpoint will be assessed using the cost per Quality-Adjusted Life-Year (QALY). The estimation of quality of life for the QALY calculation will use the quality-of-life derived EQ-5D as the main evaluation variable, and the calculation will use the area under the curve method [46]. The costs will be calculated by combining the number of treatments and cost units. The secondary economic endpoint will be assessed using effectiveness evaluation indicators such as cost per EQ-VAS calculated with EQ-VAS.

The economic analysis will be primarily evaluated based on the FAS principle and secondarily based on the PP principle to check the sensitivity of the missingness. Missing values will be imputed properly based on analysis of mechanism in the same manner as in the efficacy analysis. The first analysis period is 18 weeks (total follow-up period), and if an estimation of the subsequent period is necessary, the second analysis will be estimated by extrapolating the cost and effect after the tracking period through a regression model, or by performing decision modeling analysis. If the total analysis period (time horizon) is extrapolated after the clinical study period, the cost unit will be settled to the Korean currency unit (KRW) in 2021, and a 5% discount rate will be applied based on the economic evaluation guidelines of the Health Insurance Review and Assessment Service.

The analytical perspective of this study is a social perspective. In the baseline analysis, the representative values (such as average) of parameters will be used, and in the sensitivity analysis, probabilistic sensitivity analysis will be performed through the distribution and representative values of all possible estimated parameters.

The results will be presented in tables including the incremental cost-effectiveness ratio, cost-effectiveness plane, Cost-Effectiveness Acceptability Curve (CEAC), and value of information analysis graph. The cost-effectiveness plane will include a confidence interval confirmed by non-parametric methods. The CEAC will confirm the sensitivity of cost effectiveness according to changes in national threshold. The value of the information analysis graph will estimate the value of the information of the target population group.

When the number of samples in this study is less than 30, the statistical analysis will be performed after verifying the normality of all continuous variables through the Shapiro–Wilk test and the statistical significance level will be tested with a 0.05 *p*-value. The analysis programs are Stata (MP 16.1 Version) and R (4.3.2 Version), and extrapolation through modeling is performed using Treeage Pro (Healthcare Version 2023).

### 2.14. Data Management

The investigators will collect medical information and record it in each patient’s Case Report Form (CRF). The data will be saved confidentially in accordance with the personal information protection policy of the National Institute for Korean Medicine Development (NIKOM) and will be destroyed after the retention period of up to 10 years has ended. A separate place to store various data and records related to the conduct of this trial will be prepared with security maintained. A copy of the documents related to this trial, including patient informed consent, records of participants, and CRFs, will be kept for three years in document storage.

### 2.15. Monitoring

Monitoring oversees the progress of clinical study and regularly reviews and checks whether the study is conducted and recorded in accordance with plans, standard work guidelines, clinical research management standards, and relevant regulations. Monitoring will be conducted regularly by phone calls and visits. The monitoring staff will be composed of the A-CRO (Academic Contract Research Organization) of Daejeon Korean Medicine Hospital, which is independent of investigators and funders. The staff will periodically check and review the data storage and the progress of the trial and consults with the investigator if there is a problem. In this trial, monitoring will be conducted according to the monitoring plan, but the monitoring period and frequency can be coordinated under mutual agreement.

### 2.16. Ethics and Dissemination

This study is conducted in compliance with all applicable regulations, including the ICH GCP Guidelines, the Helsinki Declaration [47], the Korean GCP Guidelines, the Korean Pharmaceutical Affairs Act, IRB, and regulations on data protection.

It was approved by the Institutional Review Board at Daejeon University Daejeon Medical Center (DJDSKH-21-BM-04) and registered via Clinical information Services (CRIS) on 25 February 2021 (identifier: KCT0005964).

### 2.17. Study Protocol Modifications

When revising this protocol, the date, details of the revision, and reason for the revision must be reported to the IRB and approved by the IRB. The change in the study protocol will also be notified to participants.

## 3. Discussion

Hwa-Byung causes physical, social, and emotional stress not only to the patient but also to the family members who care for them [6]. Significant correlations exist between depression in wives and depression in husbands, adding a burden to family dynamics [48]. Additionally, patients often perceive their symptoms as persisting, even after apparent relief [49]. Therefore, to prevent stress from developing into a disease, it is necessary to manage stress in the daily environment using mind–body intervention techniques.

The extant research demonstrates that VR therapy provides effective symptom relief across varying time frames, thereby highlighting its potential for a lasting impact on emotional regulation disorders [50]. In the short term, VR therapy has been demonstrated to rapidly alleviate symptoms, with significant improvements achieved within just a few sessions for conditions such as anxiety with specific phobia [51]. These benefits frequently extend into the medium term, with studies indicating that emotional resilience can be sustained for up to four weeks post-treatment [52]. Furthermore, the long-term effects are promising, as research indicates a reduction in relapse rates and sustained symptom relief for up to one year following treatment [53]. The evidence suggests that VR-based ETE may provide long-lasting relief for Hwa-Byung, a condition characterized by persistent physical and emotional symptoms. Thus, VR can be positioned as a valuable complement to conventional therapeutic modalities.

This study proposes a novel approach to addressing deficits in emotion regulation, particularly in culturally bound syndromes like Hwa-Byung, by integrating virtual reality. Recognizing that symptom expression is influenced by cultural context, we designed our approach to observe participants’ symptoms through a cultural perspective, identify core emotions, and develop tailored therapeutic methods accordingly. The study’s objective is to compare the safety, effectiveness, and economic feasibility of ETE and VR-based ETE to explore their impact on the core and related symptoms of Hwa-Byung. Should the results prove meaningful, this research could contribute not only to improving public health and quality of life for patients suffering from Hwa-Byung, but also to the modernization and popularization of traditional Korean medicine through its integration with advanced digital platforms such as VR.

Moreover, the immersive nature of VR facilitates enhanced emotional processing, which has the potential to be applied to a wider range of psychiatric and psychosomatic conditions. This study contributes to the growing body of evidence supporting the role of cutting-edge technologies in advancing psychotherapeutic interventions, with the potential to improve mental health outcomes globally and pre-empt the development of a new market for digitalized mental health treatments.

## 4. Conclusions

This study proposes a novel approach to treating Hwa-Byung by leveraging the immersive capabilities of VR. By integrating VR with traditional psychotherapy methods, the VR-based ETE approach has the potential to enhance treatment outcomes for Hwa-Byung and serve as a model for treating other psychosomatic disorders. The results of this trial could significantly influence the future of psychotherapeutic interventions.

## Figures and Tables

**Figure 1 healthcare-12-02407-f001:**
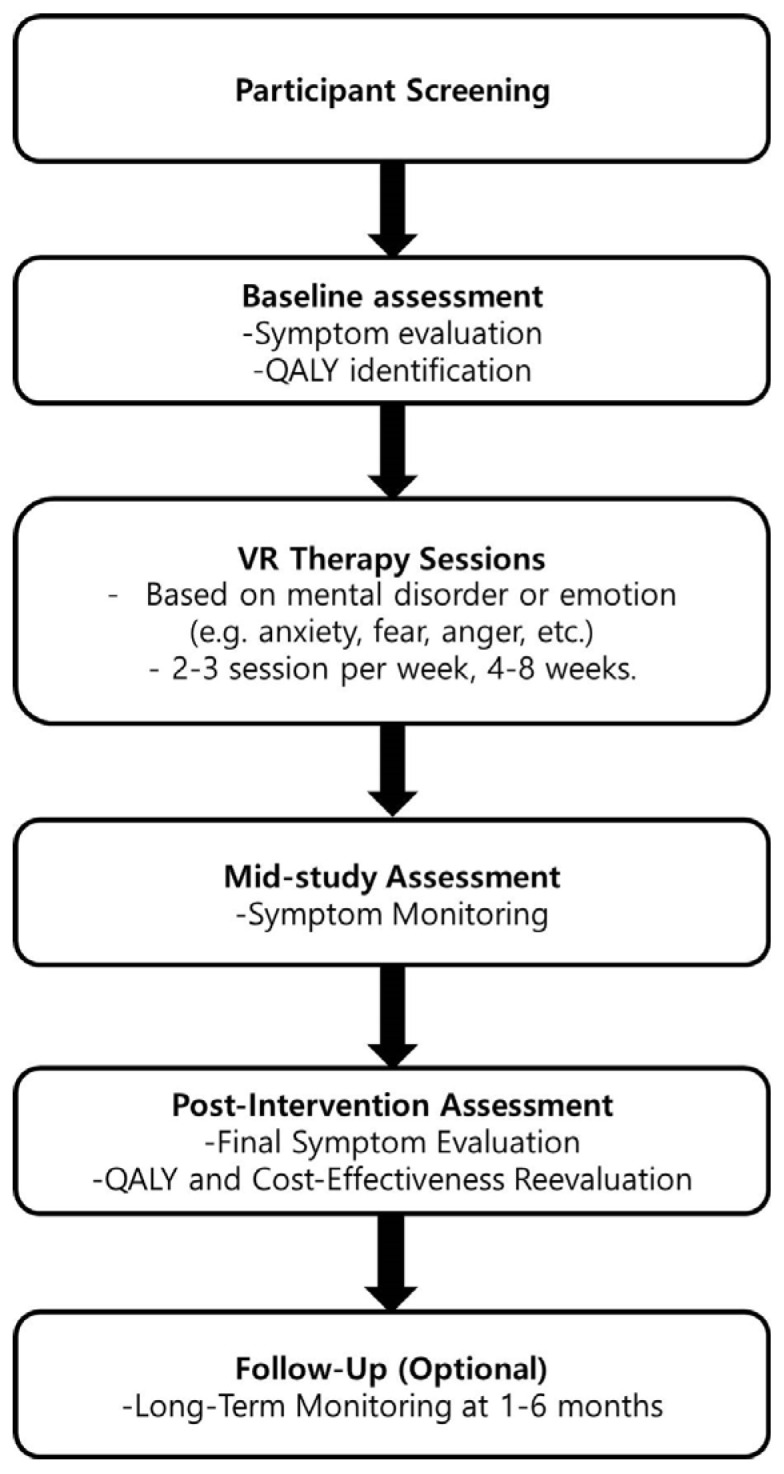
Flowchart of VR-based intervention design for psychiatric disorders.

**Figure 2 healthcare-12-02407-f002:**
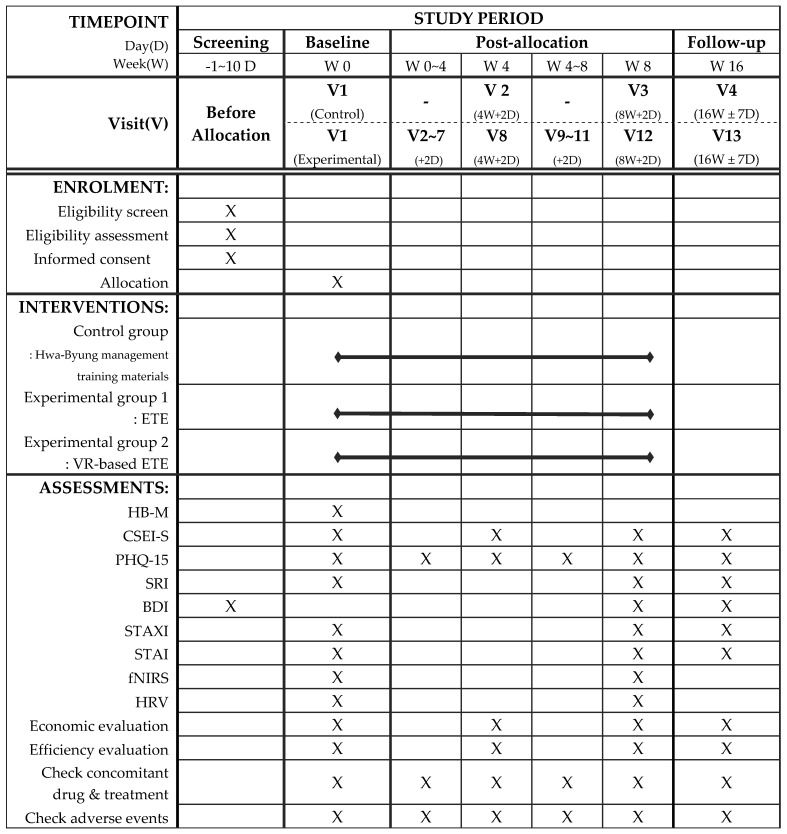
The SPIRIT schedule of enrollment, interventions, and assessments.

**Figure 3 healthcare-12-02407-f003:**
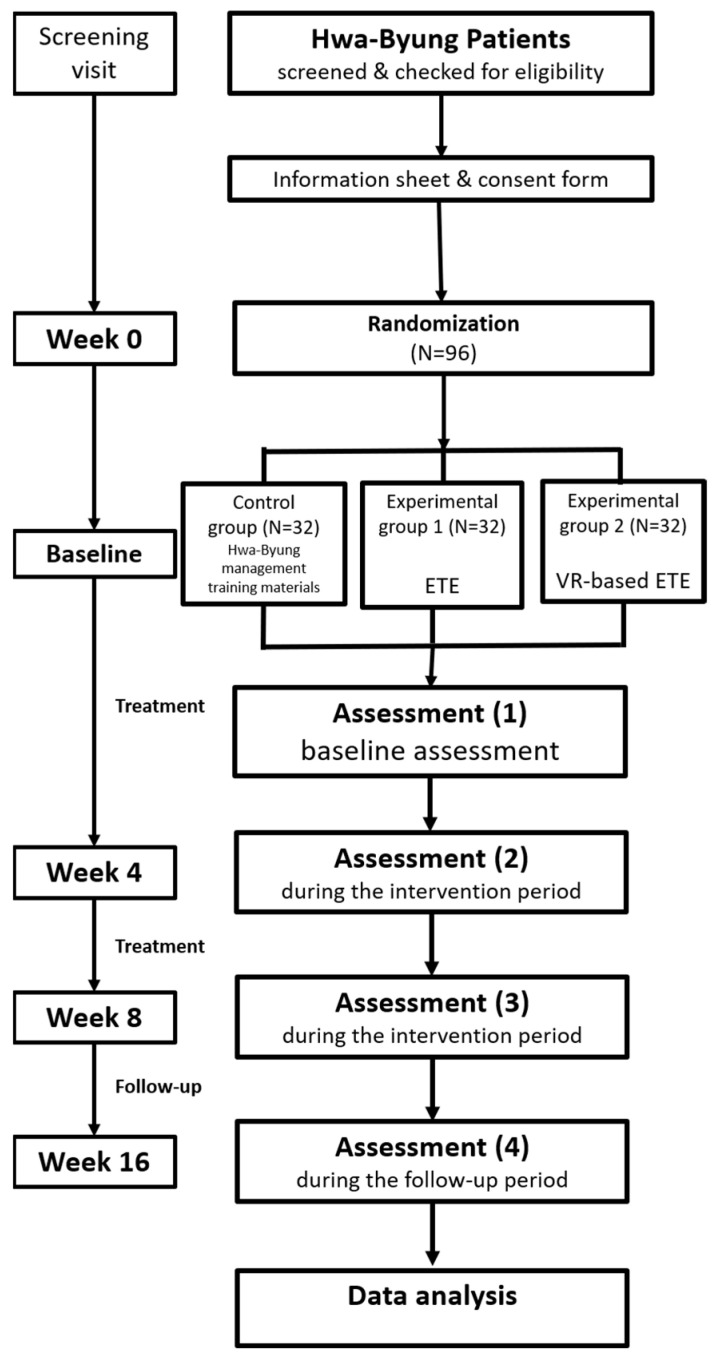
The flowchart of the study.

## Data Availability

The datasets used or analyzed during the current study are available from the corresponding author upon reasonable request.

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
