# Peer review of "Efficacy and Safety of Virtual Reality-Based Versus Traditional Emotion-to-Emotion Therapy for Treatment of Hwa-Byung: A Protocol for a Single-Center, Randomized, Assessor-Blind, Parallel-Group Clinical Trial"

_healthcare, 2024, doi:10.3390/healthcare12232407_

Round 1

Reviewer 1 Report

Comments and Suggestions for Authors

1. The article has only Introduction. It should have other theoretical parts. Introduction means that something follows. It should be more sections with separate headings. 

2. You should be more specific here and explain how will you select exclude subjects based on what concrete limits!

"The eligible subjects who meet the inclusion and exclusion criteria will be selected 111 based on a demographic survey, vital signs (blood pressure, pulse rate, and body temper-112 ature), medical history (including chief complaint, onset, motif, past history, smoking, and 113 alcohol ingestion), chest X-ray, electrocardiography (EKG), laboratory tests, Beck Depres-114 sion Inventory (BDI), and Hwa-Byung Diagnostic Interview Schedule (HBDIS) at screen-115 ing."

3. It should be better to argue how you decided to assess participants at 4, 8 and 16 weeks. Only by doubling? or there are some evidence that the remission of the problem happens between 4 and 16 weeks. The interval or period should be relevant to the healing process time. 

4. Please argue the exclusion criteria necessity for this study. 

5. The procedure should specify clearly who will perform the intervention a specialist or a resident. Are they used interchangeable or it is selected only one? IF there are more assistants, are they the same level of competence for the study and how are you going to control the expertise level of the assistence?

"ETE will be performed by a specialist in oriental neuropsychiatry or a resident who 212 has been educated by a specialist in oriental neuropsychiatry and has over one year of 213 clinical experience in neuropsychiatry of traditional Korean medicine."

6. What do you mean by that? "Drugs for transient care of other diseases will be 236 medicated after confirmation by the researcher." 

7. Is this ethical? does the researcher interfere with the medication of the patients for the sake of the research? Explain please! 

"There are no specific drugs prohibited from concomitant use, but the drugs that may 241 affect results can be prohibited at the discretion of the researchers’ decision"

8. do you have a basis for this? Based on what? Please explain!

Compliance (%) during the trial should be at least 75%. If the compliance (%) of the 284 subjects is less than 75%, it is considered poor and is excluded from the per-protocol (PP) 285 analysis.

9. "serious violations of the plan" are there mild or easy violations of the plan? 296l

Reviewer 2 Report

Comments and Suggestions for Authors

The protocol is well-designed and presents an adequate experimental scheme, your figures and tables are clearly organized, yet the manuscript needs some information to enrich and justify some decisions for therapy choice. I am recommending acceptance with major corrections that are mainly theoretical and practical specifics. Please find specific comments below.

Title. I suggest narrowing the title to be more precise and avoid over petition of “emotion-to emotion therapy”, the presentation of the title could be explained in full in the text.

Line 18. Please elaborate on what are the physical, specific manifestations of the syndrome.

The introduction needs to include more specific studies that will help future readers conceptualize Korean interventions and VR protocols.  You need to elaborate more about the specifics, trajectory, causes, social and familiar context of Hwa-Byung to justify the psychological scales. Also, the inclusion criteria. Is the condition similar between 19 and 65 years.

It is necessary to include other research that present similar analysis in different populations and countries to observe if previous research could complement or differ from yours.

Observing that you will have medical information, it will be interesting to observe the presence of a specific biomarker for improving symptoms.

Near-infrared Spectroscopy could be a useful technique, but it is not completely clear what information would provide and the justification in the background 

About the number of interventions, I suggest considering analyzing the differences between less than 12 sessions and more, to see if there are effects based on the treatment periodicity.

I notice the presence of a vast number of variables that will be obtained in the x-ray, EKG, vital signs, and laboratory tests that could give you extra information for a wider analysis, how do you consider employing them?

Line 43. Consider include other examples of emotion dysregulation.

Line 54. Please elaborate on the five elements.

The therapy description is not entirely clear, think about including a description.

Line 66.  Include VR characteristics and possible intervention design (Figure or flowchart)

Line 82. Observing the importance of traditional Korean medicine psychotherapy, it would be relevant to mention their general characteristics and treatments for psychological conditions.

Line 218. Please specify what the criteria for selection of the exposure of mindfulness mediation are.

Line 227. It is necessary to include the specifics of computer, software, and VR gear.

Line 241. Prohibited drugs. This can be an important issue in the formation of your groups. How are you considering the observation of effects of the drugs in the results?

Line 334. QALY should be observed as a variable and needs to be integrated in the introduction, considering what could be an ideal cost and if other country has implemented a similar health service.

About Korean health services. How is the attention of the general population? Does everyone be entitled for an attention of those characteristics?

Line 416. A specific context is necessary about the success of VR and the lasting effects in short, medium and long term compared to traditional approaches.

Round 2

Reviewer 1 Report

Comments and Suggestions for Authors

The authors answered all points requested. They did major changes.

Reviewer 2 Report

Comments and Suggestions for Authors

Dear Authors, 

The comments have been addressed including specific information for the sections mentioned previously. Thank you. 
